# CO$_2$ and Diode Lasers vs. Conventional Surgery in the Disinclusion of Palatally Impacted Canines: A Randomized Controlled Trial

Alessandra Impellizzeri, Martina Horodynski *, Gaspare Palaia ![ID], Gerardo La Monaca ![ID], Daniele Pergolini ![ID], Antonella Polimeni ![ID], Umberto Romeo ![ID] and Gabriella Galluccio ![ID]

Department of Oral and Maxillofacial Sciences, "Sapienza" University of Rome, Via Caserta 6, 00161 Rome, Italy
* Correspondence: martina.horodynski@uniroma1.it; Tel.: +39-3401619639

**Abstract:** Background: The aim of this RCT is to show the effectiveness of laser technology for the exposure of palatally impacted canines, using a CO$_2$ or diode laser, and to evaluate the possible bio-stimulation effect of the laser on the spontaneous eruption of the canine. Methods: This study was carried out on a sample of 27 patients, divided randomly into three groups: treated with a CO$_2$ laser (Group A), treated with a diode laser (Group B), and treated with a cold blade (Group C). Monitoring was performed at 1, 8, and 16 weeks after surgery, through photo and digital scans performed with a CS3500 intraoral scanner. Results: It was found that the average total eruptions are 4.55 mm for Group A, 5.36 mm for Group B, and 3.01 mm for Group C. The difference in eruption between groups A and B is not significant. Comparing the laser groups with the control group, it has emerged that the difference in eruption is statistically significant. Conclusion: A significant tooth movement was observed in both Groups A and B. The response of the canine to the bio-stimulation of the laser can be considered effective, resulting in a statistically significant difference between the study groups and the control group. Both lasers have the same bio-stimulatory action on the eruption of canines.

**Keywords:** impacted canines; orthodontics; intraoral scanner; digital monitoring; diode laser; CO$_2$ laser; palatally impacted canine





## 1. Introduction

The permanent maxillary canine is the most impacted tooth after the third molars, with an incidence of 0.9% to 2.2% [1–3]. It shows a predilection for the palatal over the labial side, it is more common in female patients than in male patients and it has a definite tendency toward being unilateral rather than bilateral [4].

Over the years, numerous researchers have focused on identifying the cause of impacted upper canines. The etiology of palatally impacted canines has been suggested as polygenic and multifactorial. There are two main theories: the genetic theory and the guidance theory. According to the genetic theory, palatal displacement of the canine is just another associated genetic characteristic. According to the guidance theory of canine impaction, these factors create a genetically determined environment in which the developing canine is deprived of its guidance, thus influencing it to adopt an abnormal eruption path [5].

The therapeutic approach involves surgical exposure of the impacted canines, followed by orthodontic traction to guide and align the tooth to the dental arch. For palatally impacted canines, different surgical approaches and various orthodontic traction techniques have been described in the literature [6]. There are two basic types of surgery for the exposure of a palatally impacted canine: the open and the closed technique.

The open technique includes the surgical exposure of the crown by complete removal of the bone and soft tissue directly overlying the impacted canine. Afterwards, a surgical pack might be used to cover the wound, while the canine can be either left to spontaneously erupt or an orthodontic attachment can be directly bonded on the canine in order to directly

apply traction. The closed technique, on the other hand, involves raising a full mucoperiostal flap, exposing the canine crown, and bonding an attachment to it. Afterwards, the flap is repositioned, and orthodontic traction is applied after initial healing until the canine erupts in the oral cavity and is subsequently guided to the dental arch. Although both approaches are versatile, can be adapted to each case and have been used extensively for many years, reports about their comparative performance are mixed. Several studies have evaluated various aspects of their performance, including surgical duration and postoperative recovery time, postoperative pain, periodontal health and esthetic appearance [5–7].

Today, laser technology has achieved a more important role in dentistry, and in the last two decades, there has been an increase in research studies into laser applications [8–12].

The use of a laser offers several advantages when compared with conventional surgery, including precision, minimal intraoperative hemorrhage, sterilization of the surgical area, healing with minimal scarring and decreased postoperative pain and swelling [9].

The purpose of this study is to introduce the use of a laser to expose the crown of a palatally impacted canine, avoiding the application of an orthodontic device to traction it. At the base of this research, there is an evaluation of the literature, which reports significant studies demonstrating the biostimulation effect of the photobiomodulation therapy (PBMT) [10–12].

Biostimulation means the activation of regenerative and healing processes that depends on the ability of subcellular photoreceptors to respond to visible red and near-infrared wavelengths. Stimulation of these receptors influences the electron transport chain, the respiratory chain and oxidation, expressed as an increase in the cellular metabolic processes [12,13].

Several studies have examined the effects of low-level laser therapy, known as photobiomodulation therapy, on bone formation in vitro and in vivo [12].

Photobiomodulation is one of the most promising approaches today to stimulate dental movements. It has been found that laser light stimulates the proliferation of osteoclasts, osteoblasts and fibroblasts and thereby affects bone remodeling and accelerates tooth movement. The mechanism involved in the acceleration of tooth movement is the production of ATP and the activation of cytochrome C, and low-energy laser irradiation enhanced the velocity of tooth movement via RANK/RANKL and the macrophage-colony-stimulating factor and its receptor expression [9].

Furthermore, the literature reports that high-intensity laser therapies (HILTs), the "surgical lasers", exerting a cutting action on the soft tissues [14–18], can also be used for opercolectomy, removing the soft tissue that overlies the impacted tooth.

Soft-tissue lasers have numerous applications in orthodontics, including gingivectomy, frenectomy, operculectomy, papilla flattening, uncovering temporary anchorage devices, ablation of aphthous ulcerations, exposure of impacted teeth and even tooth whitening. As an adjunctive procedure, laser surgery has helped many orthodontists to enhance the design of a patient's smile and improve treatment efficacy [17].

However, no study has evaluated the eruptive stimulating action when a laser is applied to expose the crown of impacted canines.

The aim of this RCT is to show the effectiveness of laser technology for the exposure of palatally impacted canines using a $CO_2$ laser device (Smart US20D®, DEKA—Florence, Italy) and a diode laser device (Raffaello, DMT, Lissone, Italy, 980 nm +645 nm), and to evaluate the possible biostimulation of the laser on the spontaneous eruption movement of the canine, without orthodontic traction application. Moreover, the purpose of this study is to show the accuracy of the monitoring of the movement of the impacted tooth after exposure with a laser approach using digital technologies.

## 2. Material and Methods

### 2.1. Study Design

This study was carried out on a sample of 27 subjects referred to the Orthodontics UOC of the Department of Oral and Maxillo-Facial Sciences of "Sapienza" University of Rome. A total of 15 patients presented unilaterally impacted canines, and bilateral impactions were present in the remaining 12 patients, leaving a total of 39 impacted canines. The period of

recruitment was 12 months. All the patients were informed about the content of the study, treatment methods and potential risks and benefits before providing written informed consent to take part in this study. The study received approval from the Ethical Committee of Sapienza University of Rome (#4389) and was registered in the international public register.

A preliminary investigation was performed to estimate the power of the study (PS) and to establish the effect size (ES) of the population sampled for the experimental study. Statistical significance was calculated to determine the exact number of patients needed for the study.

Assuming we want to estimate the prevalence of a disease (impacted canines) in a population, through the study of the sample, we want an estimate of the prevalence with a certain precision and a chosen level of confidence. The size can be calculated, with a 95% confidence level, using the following Formula (1):

$$n = \frac{1.96^2 \, P_{att} \, (1 - P_{att})}{D^2} \tag{1}$$

with n = sample size, $P_{att}$ = prevalence estimating and $D^2$ = absolute precision.

In our case, with a 95% confidence level and these data, when

$P_{att}$: 96.5% (0.965)
D: 10% (0.1)
then n = $(1.96^2 \times 0.965) \times (1 - 0.965)/0.1^2$ = 12.97,
leaving a value of 13.

If the recommended sample size was >5% compared to the population from which it is extracted, its sample size could be reduced; with statistical analysis, it was found that the significant sample size for this study was 13 canines.

The inclusion criteria considered in this study design were:

- Patients with palatally impacted canines;
- Male or female;
- Aged between 14 and 24 years;
- Patients reliable for follow-up;
- Patients that understand the protocol and can give informed consent.

The exclusion criteria were:

- Non-cooperative subjects;
- Inoperable subjects;
- Vestibular impacted canines;
- Systemic pathologies;
- Subject in drugs therapy.

Patients who did not meet all the inclusion criteria were excluded from the RCT. The age of the selected patients is between 14 and 24 years. Patient characteristics that were collected are age, sex and number and location of the impacted canines.

The 27 selected subjects were randomly divided into three groups, those treated with two different laser devices ($CO_2$ or diode laser) and a control group treated with a cold blade, to compare the results obtained with the gold standard method and demonstrate the effectiveness of laser technology for the exposure of palatally impacted canines.

Experimental groups:

- GROUP A (9 patients; 13 impacted canines): treated with a $CO_2$ laser (SmartXide®, DEKA, Florence, Italy, 10,600 nm) with a power of 4.5 watts in superpulsed mode (frequency: 80 Hz, fluency: 44.78 J/cm$^2$, spot diameter: 400 μm).
- GROUP B (9 patients; 13 impacted canines): treated with a diode laser (Raffaello, DMT, Lissone, Italy, 980 nm + 645 nm) with a power of 4 watts in continuous mode (fluence: from 0.1 J/cm$^2$ to 10 J/cm$^2$).

Control group:

- GROUP C (9 patients; 13 impacted canines): treated with a traditional surgical-orthodontic approach with a cold blade.

### 2.2. Analysis of Radiographs

An RX orthopantomography and a CBCT were requested from patients at the beginning of therapy (Figure 1) to accurately evaluate the cases before surgery. An evaluation of the prognosis of impacted canines was performed on OPT by two orthodontists, according to Ericson and Kurol [2,19].

The major axis of the canine and its perpendicularity to the alveolar margin, which represents the axis of the canine in its presumed ideal position, were plotted on the OPT. The examination of the CT allowed us to evaluate the three-dimensional morphology of the impacted tooth, its location and inclination in the three planes of space, the depth and the type of inclusion, and the relationships with the other elements.

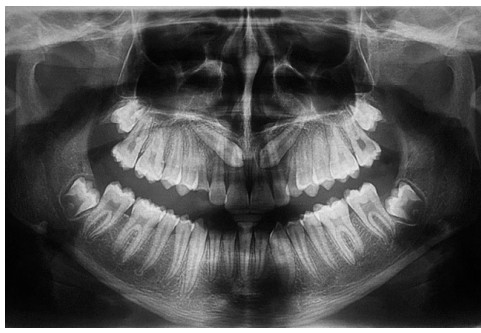 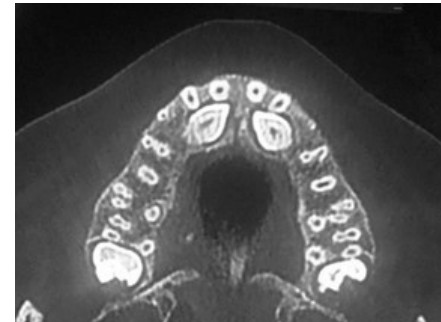

**Figure 1.** An RX orthopantomography (**left**) and a CBCT (**right**) of a patient with two palatally impacted canines.

### 2.3. Steps of the Surgery

The surgery protocol includes local anesthesia given with a Mepivacaine 2% solution with Adrenaline 1:100,000 for Groups A and C, and with Mepivacaine 3% without adrenaline, 1.8 mL solution for injection (Pierrel Spa, Milan, Italy), for Group B.

For Group A, the operculum was performed with a $CO_2$ laser, whose tip is used near to the tissue surface during the rotary movement that is necessary to expose the crown of the impacted tooth (Figure 2).

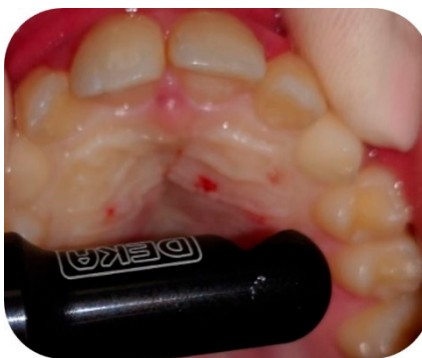 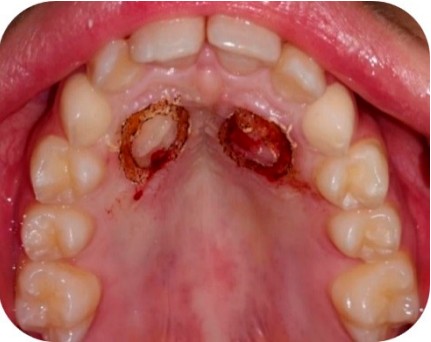

**Figure 2.** The tip of the $CO_2$ laser is used near to the tissue surface (**left**); Operculum performed with CO2 laser (**right**).

For Group B, the operculum was performed with the diode laser, whose tip is used in contact with the tissue surface during the rotary movement that is necessary to open an operculum in correspondence with the crown of the impacted tooth (Figure 3).

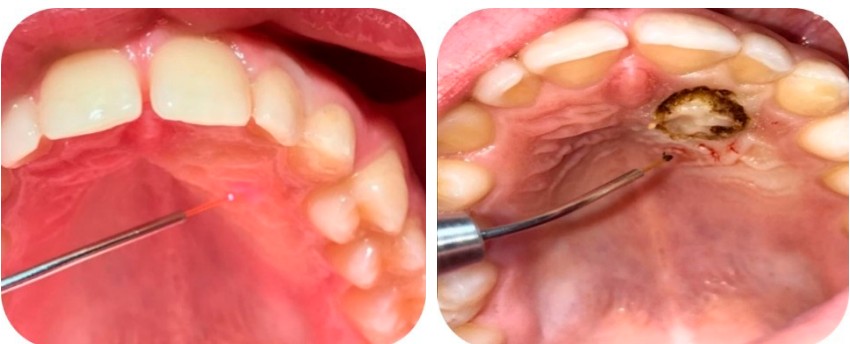

**Figure 3.** Tip of the diode laser is used in contact with the tissue surface. (**left**); Operculum performed with diode laser (**right**).

The palatal mucosa overlying the tooth is then detached with a Prichard periosteal elevator and removed by the laser's cutting action (Figure 4).

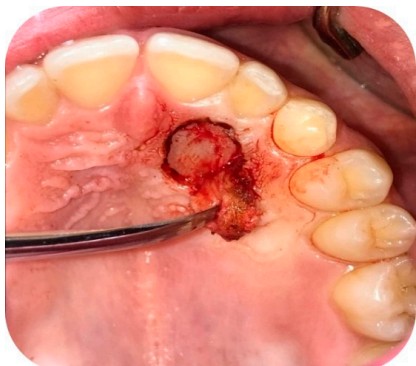

**Figure 4.** Exposure of the palatally impacted canine with a Prichard periosteal elevator.

The control group was treated with traditional cold blade surgery. An operculum is performed with the scalpel to expose the canine crown, similarly to the laser groups A and B. The use of means of traction was also avoided in this group to compare, under equal conditions, the results obtained from the three groups (A, B and C).

Afterwards, since the impacted canine was covered by bone, an osteotomy was made through a handpiece at low speed and under abundant irrigation, with a rosette bur by ISO 018 diameter (Maillefer®, Ecublens, Switzerland). The drill was used with a tangential sliding movement to the bone tissue to gradually remove it until the canine's surface is covered, which must not be damaged (Figure 5).

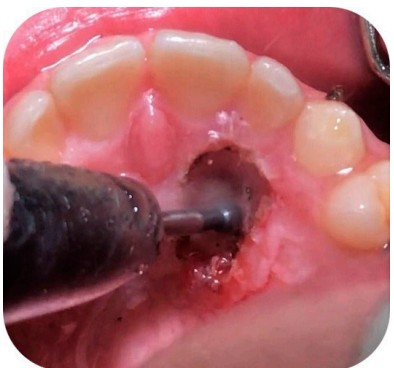

**Figure 5.** Osteotomy through a handpiece at low speed with a rosette bur by ISO 018 diameter (Maillefer®, Ecublens, Switzerland).

In Groups A and B, the lasers were applied again on the tissue surrounding the canine to use the possible stimulating effect of laser light on the periodontal ligament, which activates and speeds up tooth movement.

Finally, a periodontal dressing was applied (Coe-Pak®, GC, Tokyo, Japan), to protect the treated area of the palate (Figure 6), for about 7 days, blocked by a suture point in Vicryl 3.0 (Ethicon® V311H 3/0 SH-1 70CM, J&J Medical Devices, Somerville, MA, USA).

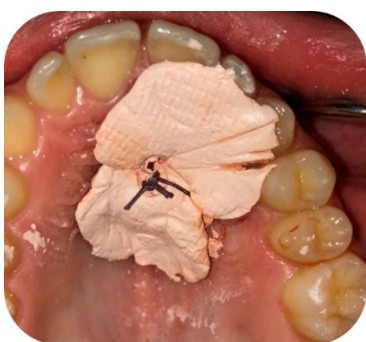

**Figure 6.** Periodontal dressing application (Coe-Pak) blocked by a suture point in Vicryl 3.0.

After 1 week, the periodontal dressing and the suture points were removed. When the tooth has erupted sufficiently into the palate, ideally in a period of four months (16 weeks), a bracket or an orthodontic button is placed on the tooth and the upper arch is bandaged with Damon System technique. In this way, the canine can be gradually guided into the dental arch.

### 2.4. Monitoring of Dental Movements

Patients underwent three check-ups at 1 week (Figure 7 left), 8 weeks (Figure 7 middle), and 16 weeks (Figure 7 right) after surgery, to evaluate and monitor, through photographic documentation and digitally colored scans through use of the intraoral scanner CS3500 (CS3500®, Carestream Health, Rochester, USA), the impacted canine's eruption or the possible early closure of the mucosa and lack of appearance of the element in the palate.

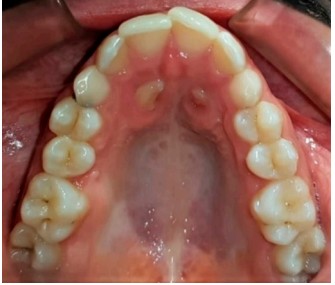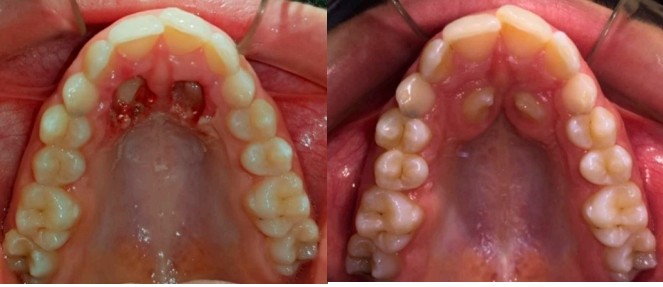

**Figure 7.** Intraoral photo at 1 week (**left**), 8 weeks (**middle**) and 16 weeks (**right**) after $CO_2$ laser surgery, that show the spontaneous eruption of the canines on the palate.

Moreover, clinical examination was carried out and questions were asked to the participants regarding the occurrence of pain, swelling or bleeding, the need to take medicine and reduced functionality of the area involved; the answers to the questions were all negative. The Visual Analog Scale (VAS) was submitted to the subjects of Groups A and B and none of them registered pain on this scale, contrary to the control group, who recorded post-operative pain in 44.4% of the cases.

The hypothesis of this work was that palatally impacted canines, after laser surgery, will undergo a "spontaneous" eruption for reactivation of the physiological eruption, without any orthodontic traction.

To obtain an accurate initial idea of the situation, Invesalius 3.1 software was used to extrapolate an STL file (T0) from each CT, which could be superimposed with the intraoral scans of the same patient at 8 weeks (T1) and 16 weeks (T2) after the surgery.

The 3D impressions (STL files) were imported into the open-source software MeshLab® (Visual Computing Lab, Pisa, Italy). After importing and superimposing the three scans at T0, T1 and T2, through the "measuring tool", it was possible to measure the distances between the cusp of the canine at T0 and the cusp of the canine that erupted on the palate at T1 and T2 (eruption values in millimeters) (Figure 8).

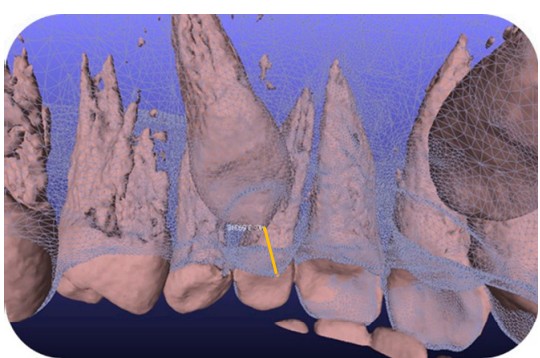

**Figure 8.** Measurement of the eruption values (distances between the cusp of the canine at T0 and the cusp of the canine erupted on the palate at T1 and T2) in mm.

The partial eruptions, obtained from T0 to T1 and from T1 to T2, were measured by superimposing the scans and calculating the distances between the initial position of the cusp of the canine and its new position at T1 and T2.

The maximum eruption of the canines was measured by calculating the distance in mm between the cusp of the canine at T0 and the cusp at T2 on the superimposed models.

When the teeth had erupted sufficiently into the palate, in an ideal period of 16 weeks, an orthodontic button was placed on the vestibular surface of the canines, the upper arch was bandaged with a self-ligating device [20,21] and the teeth could be gradually translated into the dental arch.

### 2.5. Statistical Analysis

The eruption values of Groups A and B were compared with the application of the *t*-test (also called the Student's *t*-test), to evaluate the difference between the surgery performed with the $CO_2$ and diode lasers.

The results obtained in the experimental Groups A and B were compared with the eruption values of the control Group C with the One-Way ANOVA test to evaluate the effectiveness of the surgery performed with the laser with respect to the conventional surgery performed with the cold blade.

## 3. Results

### 3.1. Prognosis of the Canines

The prognosis of each impacted tooth, according to the analysis of Ericson and Kurol, was evaluated by the OPT. The prognosis can be considered favorable when the angle "α", determined by the major axis of the impacted canine with respect to the axis perpendicular to the occlusal plane, is equal to or less than 30° if mesio-inclinated, or 45° if it is disto-inclined.

We also considered the height of the crown of the canine relative to the roots of the contiguous elements and the mesio-distal position of the cusp of the canine, which could be located in the four sectors.

It emerged that:

- A total of 16 of the 39 canines considered in the study had a positive prognosis, and the other 23 had a negative prognosis (22 canines with angle $\alpha > 30°$, 1 canine with $\alpha = 30°$);
- A total of 18 canines were in sectors I-II and 21 canines were in sectors III and IV;
- A total of 5 canines were located in the third apical of the incisor's root, 6 in the third coronal, and the remaining 28 in the third medium;
- A total of 26 canines were covered by bone and the other 13 were in mucosal inclusion only.

### 3.2. Eruption Values

The millimetric values of spontaneous eruption of the palatally impacted canines treated by laser or cold blade surgery were obtained through superimposition (with the MeshLab software) of the scans performed with the intraoral scanner CS3500.

Table 1 contains all the eruption data calculated for canines in T0-T1 (corresponding to eruption from CT to 8 weeks) and in T1-T2 (from 8 to 16 weeks), and the total eruption movement in T0-T2 (at 16 weeks).

**Table 1.** Eruption values for Groups A, B and C in T0-T1, T1-T2, T0-T2.

| Group A | | | Group B | | | Group C | | |
|---|---|---|---|---|---|---|---|---|
| T0-T1 | T1-T2 | T0-T2 | T0-T1 | T1-T2 | T0-T2 | T0-T1 | T1-T2 | T0-T2 |
| 2.84 | 1.89 | 4.74 | 3.66 | 2.28 | 5.94 | 2.04 | 0.63 | 2.67 |
| 2.33 | 1.92 | 4.25 | 1.03 | 2.72 | 3.75 | 2.1 | 0.23 | 2.33 |
| 3.13 | 1.81 | 4.95 | 4.26 | 1.91 | 6.17 | 3.12 | 0 | 3.12 |
| 2.21 | 1.48 | 3.69 | 1.44 | 1.68 | 3.12 | 2.11 | 0 | 2.11 |
| 3.34 | 1.31 | 4.72 | 3.13 | 2.21 | 5.34 | 1.76 | 0.11 | 1.87 |
| 3.11 | 0.82 | 3.93 | 2.32 | 1.86 | 4.18 | 2.23 | 0.81 | 3.04 |
| 2.31 | 1.83 | 4.14 | 4.02 | 3.02 | 7.04 | 2.98 | 1.23 | 4.21 |
| 1.29 | 1.43 | 2.72 | 4.12 | 2.78 | 6.9 | 1.86 | 0.2 | 2.06 |
| 1.78 | 1.55 | 3.33 | 3.61 | 2.69 | 6.3 | 1.62 | 0.94 | 2.56 |
| 4.71 | 1.83 | 6.54 | 3.23 | 2.68 | 5.91 | 3.01 | 0.98 | 3.99 |
| 2.66 | 2.14 | 4.8 | 3.03 | 1.82 | 4.85 | 3.19 | 1.01 | 4.2 |
| 3.48 | 2.45 | 5.93 | 3.41 | 1.97 | 5.38 | 2.89 | 0.89 | 3.78 |
| 3.54 | 1.88 | 5.42 | 2.81 | 2.02 | 4.83 | 1.33 | 1.88 | 3.21 |

We also calculated the average eruption values at each time interval for each group.

$$\text{Average eruption } 0 - \text{T1} = \frac{\sum \text{eruptions T1}}{N} \tag{2}$$

$$\text{Average eruption T1} - \text{T2} = \frac{\sum (\text{eruption T2} - \text{eruption T1})}{N} \tag{3}$$

$$\text{Average Total eruption} = \frac{\sum \text{eruptions T2}}{N} \tag{4}$$

with N = 13.

The results for Group A are:

- The average eruption in T0-T1 = 2.83 mm;
- The average eruption in T1-T2 = 1.72 mm;
- The average total eruption = 4.55 mm.

The results for Group B are:

- The average eruption in 0-T1 = 3.08 mm;
- The average eruption in T1-T2 = 2.28 mm;
- The average total eruption = 5.36 mm.

The results for Group C are:

- The average eruption in 0-T1 = 2.33 mm;
- The average eruption in T1-T2 = 0.69 mm;
- The average total eruption = 3.01 mm.

### 3.3. Comparison between the Effect of the $CO_2$ Laser and the Diode Laser

Within the 26 canines that were surgically exposed by laser operculectomy, the two experimental groups A and B, treated with the $CO_2$ laser and the diode laser, respectively, were compared.

For Group A (n = 13 palatally impacted canines treated by $CO_2$ laser) and Group B (n = 13 palatally impacted canines treated by diode laser), the measurement of the eruption path (in millimeters) of the tooth performed over 16 weeks gave the results reported in Table 1 and Figures 9 and 10.

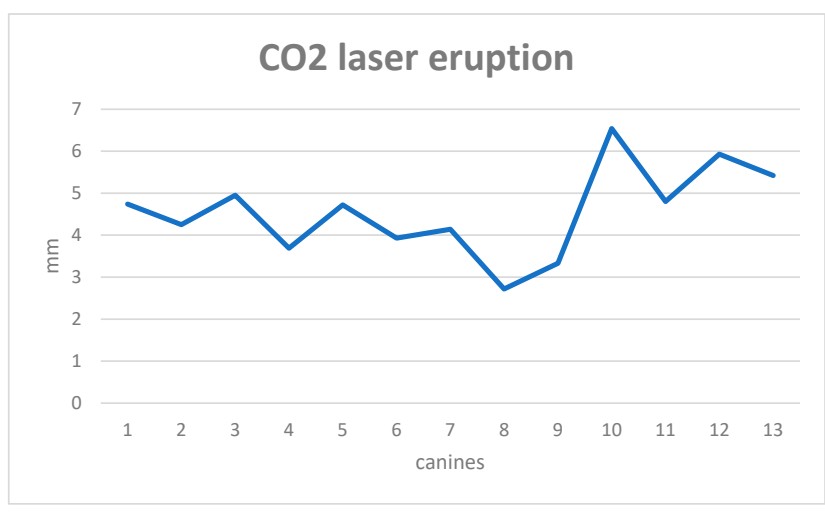

**Figure 9.** $CO_2$ laser eruption values.

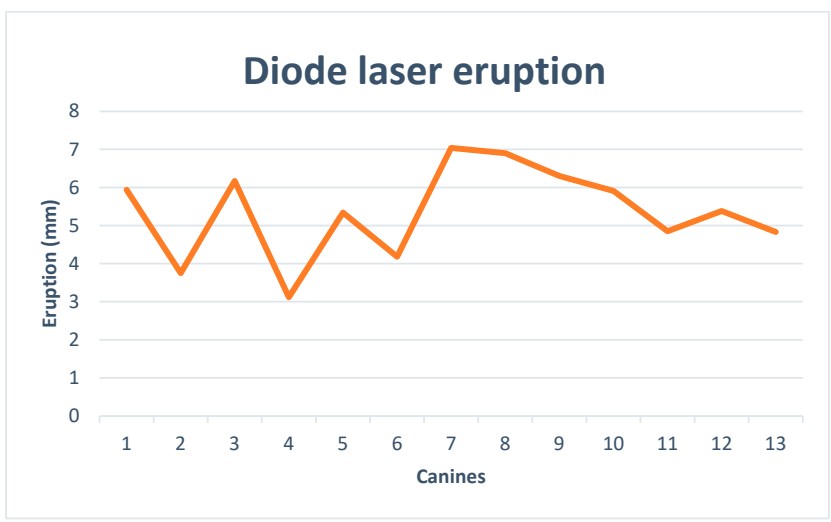

**Figure 10.** Diode laser eruption values.

To compare the two averages of the results obtained with the two different types of lasers, a *t*-test (also called the Student's *t*-Test) was applied (Tables 2 and 3).

**Table 2.** Comparison between the average eruptions obtained with the two different types of lasers.

|  | $CO_2$ **Laser** | **Diode Laser** |
|---|---|---|
| Sampling number | 13 | 13 |
| Mean | 4.55 | 5.36 |
| Standard deviation (SD) | 1.0049 | 1.140 |

**Table 3.** Comparison between the average eruptions obtained with the two different types of lasers.

| **T** | **0.0471** |
|---|---|
| Degrees of freedom | 24 |
| P (significance level) | 0.7280 |

A significance level of $\alpha = 0.05$ was chosen.

The degrees of freedom correspond to d.f. $= N - 2 = (13 + 13) - 2 = 24$, where N corresponds to the sum of the subjects belonging to the two samples N = n (group A) + n (group B).

The difference between the observed means of the two groups is not significant for $p < 0.05$, so it is not attributable to the different action of the two different types of lasers but is considered accidental.

### 3.4. Comparison between the Study Groups (Group A + Group B) and the Control Group C

The spontaneous eruptions in mm of canines of the study groups (Groups A and B; n = 26 canines disincluded by laser surgery) were compared with the eruptions of the canines of the control group (Group C; n = 13 canines treated by the conventional surgical approach).

The total eruption values of the control groups are reported in Table 1.

To compare the results obtained with the two different types of lasers with the control group, the one-way ANOVA test was applied. The results of the ANOVA test are reported in the following tables and graphics (Tables 4–6; Figures 11–13).

**Table 4.** Comparison between Groups A, B and C in T0-T1.

| ANOVA | Sum of Squares | Degrees of Freedom | Mean Square | Contribution, % | Fisher Value |
|---|---|---|---|---|---|
| Between | 3.84 | 2 | 1.92 | 12.98 | |
| Within | 25.76 | 36 | 0.72 | 87.02 | 2.69 |
| Total | 29.60 | 38 | 2.64 | | |

**Table 5.** Comparison between Groups A, B and C in T1-T2.

| ANOVA | Sum of Squares | Degree of Freedom | Mean Square | Contribution, % | Fisher Value |
|---|---|---|---|---|---|
| Between | 17.01 | 2 | 8.50 | 67.71 | |
| Within | 8.11 | 36 | 0.23 | 32.29 | 37.74 |
| Total | 25.12 | 38 | 8.73 | | |

**Table 6.** Comparison between Groups A, B and C in T0-T2.

| ANOVA | Sum of Squares | Degree of Freedom | Mean Square | Contribution, % | Fisher Value |
|---|---|---|---|---|---|
| Between | 37.07 | 2 | 18.53 | 49.18 | |
| Within | 38.30 | 36 | 1.06 | 50.82 | 17.42 |
| Total | 75.37 | 38 | 19.60 | | |

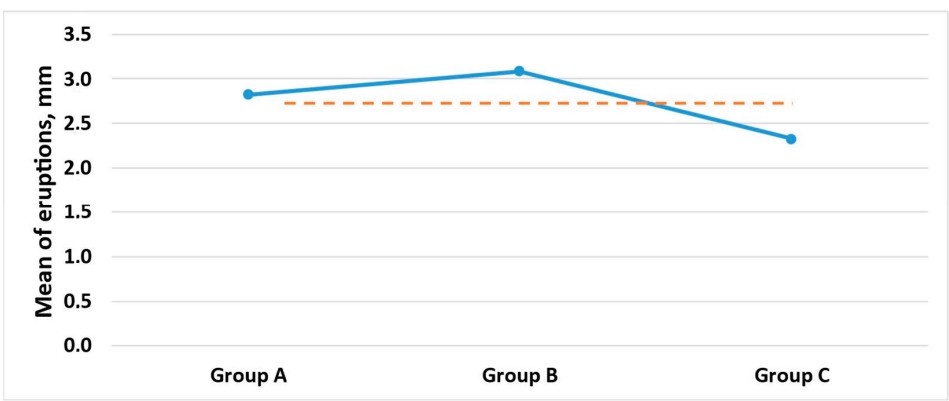

**Figure 11.** Mean of eruption in mm in the interval T0-T1.

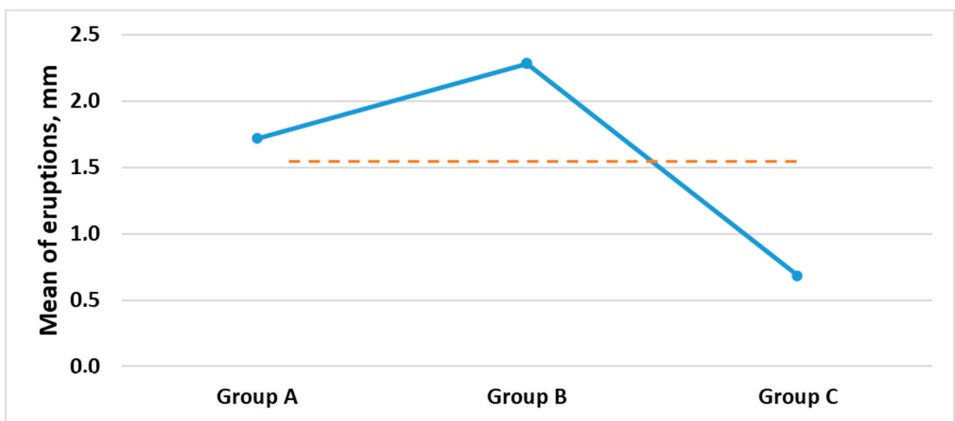

**Figure 12.** Mean of eruption in mm in the interval T1-T2.

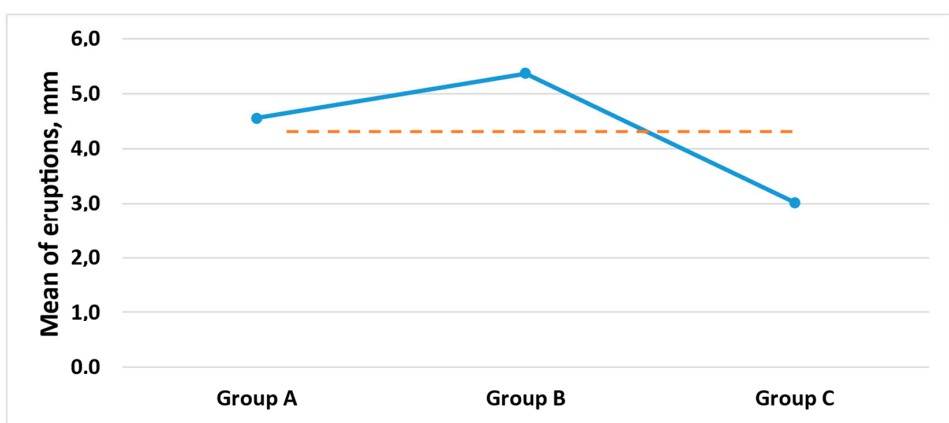

**Figure 13.** Mean of eruption in mm in the interval T0-T2.

It has emerged that the differences in the results obtained in the T1-T2 (Fisher value 37.74) and T0-T2 (Fisher value: 17.42) intervals are statistically significant, while for the T0-T1 interval, the difference is not significant.

### 3.5. Speed of Eruption

Starting from the eruption data, the average speeds in the intervals T0-T1, T1-T2 and T0-T2 were calculated, considering T0 as the value obtained from the STL file extrapolated from the CT.

The speeds in mm/day in the time ranges considered were found with the following formula:

$$Speed = \frac{eruption\ Tf - eruption\ Ti}{Tf - Ti} \tag{5}$$

with Tf = final time of the interval and Ti = initial time of the interval.

For Group A:

The average total speed is 0.038 mm/day.

For Group B:

The average total speed is 0.045 mm/day.

For Group C:

The average total speed is 0.025 mm/day.

All the data were measured by two different operators to determine the method error of digital measurements. For each compared value, the correlation coefficient, the relative and absolute errors and the mean differences were calculated (Table 7).

**Table 7.** Range of correlation, error (%) and mean difference (%) values.

| Correlation Coefficient | Error of Method | Mean Difference |
|---|---|---|
| 0.994–0.999 | 0.20–0.99% | 0.17–0.87% |

A coefficient of correlation close to 1 indicates a "positive" and "strong" relationship between the two examined variables. In other words, when increasing one variable, the other one will increase too. In this case, in particular, a high mean coefficient of correlation (0.9965) was observed, which indicates a strong correlation, suggesting less dependence on who takes the measurement itself, i.e., it is independent of the operator. This result is also supported by the absolute error values and the mean differences, which are both positive.

## 4. Discussion

In recent years, different investigators have studied the results of low-level laser therapy (LLLT) during orthodontic tooth movement. Lasers with an output energy below 500 mW are proven to have a biostimulatory effect on tissues without increasing the temperature of the treated region above the normal body temperature. So, they have the potential to accelerate tooth movement by means of influencing the remodeling of alveolar bone without unwanted impacts on the tooth and periodontium. Histologic investigations revealed that LLLT during orthodontic tooth movement can profoundly affect cell-mediated alveolar bone remodeling [22].

Bozcurt et al. in their study evaluated the effects of diode laser biostimulation on cementoblasts and found that the biostimulation setting of a diode laser modulates the behavior of cementoblasts, inducing the mineralized-tissue-associated gene's mRNA expressions and mineralization on the molar's root [23].

The literature also reports numerous applications of lasers in oral surgery (high-intensity laser), for example for operculectomy to remove the soft tissue that overlies an impacted tooth [24].

However, the bio-stimulatory action of high-intensity lasers used to expose the crown of the impacted canines has not yet been demonstrated.

Kokich et al. [25] recommended an alternative disinclusion technique with earlier timing for uncovering palatally impacted canines. They timed the uncovering before the start of orthodontic treatment, the bone over the crown was removed down to the cementum–enamel junction and an opercolum was made on the palatal mucosa with a surgical scalpel. They hypothesized that, when the bone and tissue have been removed, these canines will erupt on their own in about 6 to 8 months. This consideration suggests that when the teeth are freed from the overlying tissue, a spontaneous movement of the impacted canine is activated.

In this RCT, at the end of the monitoring of 16 weeks, a significant tooth movement was observed in both Groups A (mean eruption T0-T2: 4.55 mm) and B (mean eruption T0-T2: 5.36 mm). Furthermore, the exposure of part of the dental crown allowed, in all these cases, the application of a bracket or a button to align the tooth in the dental arch. In Group C, the values of total eruption were lower (mean eruption T0-T2: 3.01 mm), demonstrating less movement of the canine in the surgery, with respect to the two study groups. Monitoring with digital technologies has allowed us to obtain more precise measurements than the monitoring on plaster casts [26,27].

The one-way ANOVA test suggest that the response of the canine to the bio-stimulant action of the laser, applied to expose the crown, can be considered effective in the period of monitoring (T0-T2), in particular in the interval T1-T2, resulting in a statistically significant difference between the two study groups and the control group.

A further purpose of this study was the evaluation of the possible different action between the $CO_2$ laser (wavelength: 10,600 nm; power: 4.5 Watts) used in superpulsed emission mode and the diode laser (wavelength: 980 nm + 645 nm; power: 4 Watts) used in continuous wave emission mode.

Comparing the millimeters of eruption of the canines treated with the two different types of lasers and applying the Student's *t*-test, we found a superimposable value.

This means that both lasers have the same bio-stimulatory action on the eruption of canines, regardless of their wavelength, power or emission mode, and this suggests that the action of the laser itself can stimulate the spontaneous eruption of the treated tooth.

This study has some limitations. The main criticisms of the study are the small sample size and the lack of histological analysis of the tissues surrounding the impacted canines, which could further support the hypothesis that the high-intensity laser could have a stimulating action on tooth movement. It would be interesting to study this further, expanding the sample size, given the promising results of the current study.

## 5. Conclusions

The present study was conducted in order to evaluate the effectiveness of laser surgery as an alternative approach to conventional surgical–orthodontic treatment and to evaluate the possible effect of this laser on the canine's movement after the operculectomy. The results of the study are promising as it has emerged that the spontaneous eruption times of the canines treated with laser surgery were shorter than the times described in the literature [3,5], which could be an advantage attributable to the use of the laser.

The surgical exposure of the maxillary deep impacted canines by $CO_2$ and diode lasers has shown undoubted advantages: an absence of bleeding during and after the procedure, no stitches, relative ease and speed of execution, reduced or absent postoperative symptoms and decontaminating action on the treated tissues.

Furthermore, it has emerged that post-operative wound healing was rapid, and all patients treated with laser surgery reported less discomfort in the post-operative period, and only minor discomfort during speech and chewing, than patients treated with conventional surgery (control group). The young patients resumed normal school and sports activities the day after the surgery.

One week after the surgery, no pain or discomfort was reported by patients of the experimental groups.

The results obtained indicate the effectiveness of the new laser surgery approach proposed in this study, although it was performed on a small sample of patients.

**Author Contributions:** Conceptualization, A.I. and M.H.; methodology, A.I.; software, M.H.; validation A.P., U.R. and G.G.; formal analysis, A.I.; investigation, M.H.; resources, G.L.M.; data curation, D.P.; writing—original draft preparation, M.H.; writing—review and editing, M.H. and A.I.; visualization, G.L.M., A.P., U.R. amd G.G.; supervision, G.G.; project administration, G.G.; funding acquisition, U.R. and G.P. All authors have read and agreed to the published version of the manuscript.

**Funding:** This research received no external funding.

**Institutional Review Board Statement:** The study was conducted in accordance with the Declaration of Helsinki, and approved by the Ethics Committee of "Sapienza" University of Rome (#4389).

**Informed Consent Statement:** Informed consent was obtained from all subjects involved in the study.

**Data Availability Statement:** Data available on request due to ethical restrictions.

**Conflicts of Interest:** The authors declare no conflict of interest.

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
