# Peer review of "CO2 and Diode Lasers vs. Conventional Surgery in the Disinclusion of Palatally Impacted Canines: A Randomized Controlled Trial"

_photonics, doi:10.3390/photonics10030244_

Round 1
Reviewer 1 Report
The article introduces two applications of laser technology for the disinclusion of impacted canines, comparing some of the outcomes with the more conventional cold blade technique. A cohort of patients is followed in time after the 3 surgeries, comparing the discomforts experienced and, most important, measuring the movements of the released teeth in space and time.
The article presents an interesting new perspective in oral surgery, but the impression is that the findings are very limited in terms of number of patients analyzed and measurements performed. Additionally, since the journal deals with photonics, the physics behind the phenomenon and its interaction with the biology of the cells and tissues of the oral cavity is practically unexplored.
In itself the paper is complete, but I'd suggest resubmission as-it-is to a different journal, perhaps more towards dental sciences.
Some general observations:
The english form is ok, although a revision could benefit the readers (use of the word "less", series of "of" in some sentences, etc).
A part that should be modified to improve clarity of the text is paragraph 3.1, when info is given about the 39 teeth analyzed: in first point about prognosis, 16+22 is not 39 (one tooth missing). in third point about final tooth level, 5+6+30 is not 39, either.
In the same list, the last point claiming that 26 teeth were covered by bone left me very confused, because this point is not mentioned much more in the article, however the number 26 comes back several times and I am not sure if it is just a coincidence. The authors should be a bit clearer about how an impacted canine could be located in the palate (under which tissue layers) and later, for example at par. 3.3 specify that the operculectomy for 26 teeth involved removing bone. Is the final statistics all about the 26 teeth?
Reviewer 2 Report
The study “CO2 and diode laser vs conventional surgery in the disinclusion of palatally impacted canines: a randomized controlled trial” conducted in order to evaluate the effectiveness of laser surgery as an alternative approach to conventional surgical-orthodontic treatment and to evaluate the possible effect of this laser on the canine’s movement after the operculectomy, results clear in relation to the exposure of the data obtained. The results indicate the effectiveness of the new approach proposed, although, as also mentioned by the authors, it is performed on a small sample of patients. I think that the manuscript could be considered for submission to Photonics Journal, but some revisions should be done:
1)The introduction needs to be implemented;
2) Additional bibliography is required;
3) The conclusions should be written in a more discursive and articulated form, in order not to resemble a summary list.
Reviewer 3 Report
Dear authors,
the article is very interesting for the reader. It is very methodically structured. The section material and methods is well described. Some points have to be discussed:
- the section "aim of the study" should be integrated in the introduction as last point and not in the section material and methods.
- as known a CO2 laser has the absorption peak in bone and many studies show the main disadvantage of bone necrosis. The laser was used with wather cooling?
- I think figure 13 is turned upside down.
- a statistical analysis was performed and the methodology well described. Nevertheless, the patient sample size is small with n=9 per group. This also reflects the significance level, which was set at alpha=0.05. So I would rather talk about tendency, effectiveness...
-Placing so much emphasis on biostimulation I would recommend describing this effect as well. An article has recently been published that tries to describe this effect in vitro (SB Bozkurt, et al. Biostimulation with diode laser positively regulates cementoblast functions, in vitro, 2017). Can be discussed in the discussion section
- the literature is well documented, although the cited papers are rather older. Especially for biostimulation I think there are newer articles.
Round 2
Reviewer 1 Report
No additional comments
Reviewer 3 Report
Dear authors,
thanks for the corrections and the implementation of my comments. A final correction in lines 423 and 430: operculectomy and operculum.